# De-Implementation of Detrimental Feeding Practices in Childcare: Mixed Methods Evaluation of Community Partner Selected Strategies

**DOI:** 10.3390/nu14142861

**Published:** 2022-07-12

**Authors:** Taren Swindle, Julie M. Rutledge, Dong Zhang, Janna Martin, Susan L. Johnson, James P. Selig, Amy M. Yates, Daphne T. Gaulden, Geoffrey M. Curran

**Affiliations:** 1Department of Family and Preventive Medicine, University of Arkansas for Medical Sciences, Little Rock, AR 72205, USA; dzhang@uams.edu (D.Z.); jmartin@uams.edu (J.M.); dgaulden@uams.edu (D.T.G.); 2College of Applied and Natural Sciences, School of Human Ecology, Louisiana Tech University, Ruston, LA 71272, USA; rutledge@latech.edu (J.M.R.); yates@latech.edu (A.M.Y.); 3Department of Pediatrics, University of Colorado Anschutz Medical Campus, Aurora, CO 80045, USA; susan.johnson@cuanschutz.edu; 4College of Public Health, University of Arkansas for Medical Sciences, Little Rock, AR 72205, USA; jpselig@uams.edu; 5Department of Pharmacy Practice and Psychiatry, University of Arkansas for Medical Sciences, Little Rock, AR 72205, USA; currangeoffreym@uams.edu; 6Central Arkansas Veterans Healthcare System, Little Rock, AR 72205, USA

**Keywords:** de-implementation, childcare, feeding practices, nutrition, implementation science, early care and education

## Abstract

This pilot evaluated strategies to decrease detrimental feeding practices in early care and education, which are hypothesized to compete with evidence-based feeding and obesity prevention practices. This study made two key comparisons: (1) a between-site comparison of sites receiving (a) no implementation or de-implementation strategies (i.e., Basic Support; B), (b) implementation strategies only (i.e., Enhanced Support; E), and (c) implementation and de-implementation strategies (i.e., De-implementation + Enhanced Support; D + E) and (2) a within-site pre-post comparison among sites with D + E. At nutrition lessons, the D + E group had more Positive Comments (Hedege’s *g* = 0.60) and higher Role Model fidelity (Hedege’s *g* = 1.34) compared to the E group. At meals, assistant teachers in the D + E group had higher Positive Comments than in the B group (*g* = 0.72). For within-group comparisons, the D + E group decreased Negative Comments (*t*(19) = 2.842, *p* = 0.01), increased Positive Comments (*t*(20) = 2.314, *p* = 0.031), and improved use of the program mascot at nutrition lessons (*t*(21) = 3.899, *p* = 0.001). At meals, lead teachers’ Negative Comments decreased (*t*(22) = 2.73, *p* = 0.01). Qualitative data identified strengths and opportunities for iteration. Despite a COVID interruption, mid-point comparisons and qualitative feedback suggest promise of the de-implementation strategy package.

## 1. Introduction

De-implementation is a concept of increasing interest to the field of implementation science. In fact, the majority of studies on de-implementation have occurred in the last decade [1,2]. De-implementation strategies focus on reducing, removing, replacing, or restricting practices deemed as having limited value or research evidence and/or potentially causing harm [3]. To date, de-implementation research has focused primarily on clinical practices with less research in community settings (i.e., 95% of funded federal grants on de-implementation in clinical care settings) [2]. Further, although engagement of key partners is proposed to be central to de-implementation efforts [1,4], few studies illustrate the process and outcomes of collaborating with community/clinical partners to select and test de-implementation strategies. Partner engagement may be even more critical in community settings for successful de-implementation efforts given less structured systems for intervention, the influence of community and cultural norms, and the unique psychological and emotional aspects of giving up a habit (versus starting a new one) [4].

Early care and education (ECE) is an important community setting for primary prevention of disease and promotion of child health. Specifically, early childhood is a key period for laying the foundation for healthy nutrition and movement habits [5,6]. Habits established in early childhood are likely to persist across the lifespan [7,8,9]. The ECE setting can support healthy development for children through exposure to healthy foods, provision of opportunities for movement, and supporting families in healthy habits. The mealtime environment in ECE is particularly important given that children in ECE settings may consume over 500 meals and snacks with their ECE teacher each year. Evidence-based practices (EBPs) in ECE that support child health include supporting trying new foods without pressure [10,11], cueing children to their hunger and satiety [12,13,14], saying positive comments about the foods served [13,15], role modeling intake of healthy foods [13,16,17], and positive repeated exposures to healthy foods [15,18,19].

The EBPs that support child health in ECE can be undermined by practices that are counter to or compete with their use; we deem these detrimental feeding practices. Detrimental feeding practices fail to support the development of child self-regulation in eating, are coercive in nature, and/or are destructive to the mealtime environment. Examples include pressuring children to eat more [10], comparing children’s eating habits [20], hurrying children through the meal [21], discouraging food exploration [22], and using food as a reward [23]. Unfortunately, detrimental feeding practices are common. For example, recent observational data suggest that ECE teachers pressure children to eat 7 times per meal on average and up to 32 times per lunch per classroom (i.e., more than once per minute) [24]. At the same time, ECE teachers cued children to their own hunger/satiety less than once per meal, on average [24]. Detrimental feeding practices have been linked with negative outcomes for children including decreased intake of healthy foods [10], increased intake of unhealthy foods [25,26], food aversions [27], neophobia [25], emotional eating [23], picky eating [23], diminished self-regulation [28], and excess weight development [23].

The determinants of detrimental feeding practices in ECE contexts are complex. Prior research has identified both contextual and individual factors that influence feeding practices. Contextual factors include mealtime location/pace, food choices, training in mealtime and feeding practices, and mealtime policies [29,30,31]. Individual-level factors include personal preferences/experiences, self-efficacy, and beliefs related to food and mealtime [30,31,32,33]. A successful effort to de-implement detrimental feeding practices would require attention to these factors as well as locally salient influences on feeding practices. The complexity of determinants of teacher feeding practices, their potential for basis in cultural origins and norms, and the well-intended nature of practices to serve the needs of children (e.g., pressure to eat in a food insecure environment) [32] suggest the need for community partner-selected strategies to reduce, remove, and/or replace detrimental feeding practices [4].

The current study focused on developing and testing strategies to decrease detrimental feeding practices, which were hypothesized to compete with positive feeding practices and other evidence-based obesity prevention practices. That is, we expected that reducing detrimental practices would create space for (i.e., be associated with) an increase in positive feeding and evidence-based obesity prevention practices. These shifts would have the potential to support child health outcomes in the long term (e.g., prevent excess weight and increase FV intake). We examined for shifts in feeding and obesity prevention practices across time in a group receiving de-implementation and implementation strategies. We also examined between-group differences among groups receiving no implementation strategies, implementation strategy support only, and both implementation and de-implementation strategy support. Differences were examined at both meals and lessons of a nutrition promotion intervention called WISE (Together, We Inspire Smart Eating).

## 2. Methods

### 2.1. Study Design

Per our published protocol [34], this study makes two key comparisons: (1) a within-site pre-post comparison assessing changes in detrimental feeding practices and evidence-based feeding and obesity prevention practices among sites receiving the de-implementation strategy and (2) a between-site comparison of sites receiving (a) no implementation or de-implementation strategies (i.e., Basic Support; B), (b) implementation strategies only (i.e., Enhanced Support; E), and (c) implementation and de-implementation strategies (i.e., De-implementation + Enhanced Support; D + E). The sites receiving B were a part of a small-scale Hybrid Type III randomized trial; additional detail on that trial, the specification of the implementation strategies, and its findings are published elsewhere [Blinded]. All study activities were reviewed and approved by the internal review board at the University of Arkansas for Medical Sciences. Teachers participating in interviews provided verbal consent, and data collection activities in the classroom were deemed consistent with usual educational practice by the IRB. Teachers received USD5 incentives for completion of monthly surveys and interviews.

### 2.2. Participants

The project was conducted in two southern states of the US. D + E sites (*n* = 3) resided in one state and included one Head Start agency with two sites (*n* = 6 teachers and 3 classrooms, *n* = 26 teachers and 13 classrooms) and one publicly funded preschool center (*n* = 20 teachers and 10 classrooms). In the second state, 4 sites were in the E group (*n* = 20 classrooms, 39 teachers); 5 sites were in the B group (*n* = 18 classrooms, 36 teachers).

### 2.3. Intervention

WISE is a nutrition promotion and obesity prevention program designed to increase eating self-regulation and consumption of fruits and vegetables [35,36,37]. The evidence-based WISE components [35] include (1) multiple hands-on exposures to fruits and vegetables (Hands-On); (2) use of a mascot puppet to promote fruits and vegetables to children (Mascot Use); (3) appropriate role modeling by ECEs (Role Modeling); and (4) positive ECE feeding practices (Positive Comments). Details of WISE can be found in previous reports [35,37]. Briefly, WISE lessons occur during classroom instruction time, and the WISE training encourages ECEs to use WISE components 2 through 4 at meals as appropriate. Detrimental feeding practices (Negative Comments) are counter to WISE components.

### 2.4. De-Implementation Strategy

The development of our de-implementation strategy was guided by the Niven model of de-implementation [1], salient theoretical domains in behavior-change theories [38], and the categories of possible implementation strategies [39]. Partner engagement is central to the Niven model and was operationalized through the application of Evidence Based Quality Improvement (EBQI) methods in our study. The EBQI panel consisted of both Head Start and publicly funded preschool center teachers (*n* = 8), directors/administrators (*n* = 2), mentor Head Start teacher panelists from one state who had participated in an EBQI process prior (*n* = 2), along with parents of children (*n* = 2) from both the Head Start and publicly funded preschool program. To recruit community partners, the research team shared information on the study with sites and invited volunteers. Directors and teachers volunteered based on these invitations. Directors nominated parents for the panel, suggesting parents that they believed would be active and engaged participants. Directors collected consent to contact from parents, and the research team contacted the parents to discuss participation. Community partners received USD50 for participation in Evidence-Based Quality Improvement (EBQI) meetings. 

EBQI sessions lasted 2 h and covered the following topics. *In EBQI session 1*, the research team presented a summary of qualitative data on the determinants of feeding practices in ECE, conducted a “member checking” exercise with participants to check the validity of this summary, and reached consensus on key barriers and facilitators to drive selection of the de-implementation strategies. *In EBQI session 2*, the research team presented potential de-implementation strategies mapped by the research team to the Expert Recommendations for Implementing Change (ERIC) [40] taxonomy of implementation strategies with consideration of the theoretical domains of behavior change [38]. To reach a consensus on the implementation strategies, we used concept mapping [41]. In this approach, community partners rated each potential strategy on its relative importance and feasibility on a scale of 1 (low importance/feasibility) to 10 (high importance/feasibility). Partners’ ratings were captured with REDCap [42], and data were processed in real time to generate a Go-Zone plot. Strategies rated as both highly important and highly feasible were prioritized and operationalized through discussion. *In EBQI session 3*, we presented the preliminary plans and draft materials related to prioritized strategies and collected feedback for revisions. *In EBQI session 4*, we launched a pre-test of the materials in the classrooms in which the teachers taught to take place between EBQI sessions. *In EBQI session 5*, we gathered feedback from community partners about the feasibility and acceptability of the de-implementation strategy to inform iterations and improvements to the approach. These sessions took place over 6 months in the school year prior to the D+ E implementation, with approximately one month between each session.

Table 1 presents the specification of the resulting de-implementation strategy package consistent with recommendations from Proctor et al. [43] in implementation strategy reporting. First, teachers participated in a dynamic training, led by professional external improvisational trainers, and driven by improvisation methods to illustrate and engage teachers about the positive effects of desired feeding practices (e.g., autonomy granting) and the negative consequences of inappropriate feeding practices (e.g., focus on short-term compliance, pressuring children) [44]. Next, teachers selected goals in two areas: (1) a feeding practice they wanted to stop or reduce in their classroom (e.g., pressuring children to eat more, comparing children’s feeding practices, hurrying to finish) and (2) a feeding practice they wanted to start or increase (e.g., offering positive comments, encouraging food exploration, cuing children to hunger/satiety). The teachers selected their 2 goals from a menu of 5 “stop” and 5 “start” options. Then, these goals were discussed with their co-teacher and a peer teacher using prompts to explore potential barriers and to increase mutual accountability (i.e., peer learning collaborative). Finally, the teachers selected the type of support they wanted to achieve their goals, choosing from (1) environmental reminders (i.e., classroom posters), (2) expert recommendations and research evidence, (3) how-to educational resources (e.g., videos, step-by-step guides), and (4) personalized audit and feedback. The program was designed to focus on a *self-selected* goal for the fall semester and a *researcher-suggested* goal for the spring semester.

### 2.5. Measures

Data were collected in alignment with our published protocol [34] until interruption by COVID-19. Specifically, we collected qualitative interviews as indicators of our primary outcomes of feasibility and acceptability, which took place by phone after COVID-19 closures. Study staff collected classroom-based measures of secondary outcomes of feeding practices and fidelity to nutrition promotion practices prior to intervention (baseline) and during the winter of the school year (follow-up), approximately 5 months after de-implementation training. Child outcome data were collected at baseline, but follow-up data for comparison could not be collected. WISE coaches (i.e., implementation facilitators) logged all activities and delivery of strategies used in a REDCap [42] database as described in a prior study [45].

*Table Talk-Revised (TT-R).* To assess de-implementation of detrimental feeding practices and implementation of evidence-based feeding practices at both lessons and meals, we used the Table Talk-Revised (TT-R) tool, which is aligned with the feeding practice goals selected by teacher. The TT-R is designed to provide an in-person, observational assessment of ECE teacher’s statements at mealtimes [24]. Trained observers collected TT-R data after demonstrating 85% reliability or better with a gold-standard observer on recorded and field observations. The TT-R is designed to collect data on the occurrence of specific ECE teacher feeding behaviors with a floor of zero and no celling. This measurement approach allows the TT-R to be maximally representative of the feeding environment and more sensitive to change (versus a capped measure). Total scores for positive (e.g., shares positive comments about the food) and negative (e.g., pressures child to eat) communications were created by totaling items in the corresponding category after the observation for each ECE teacher. That is, scores represented herein represent the total number of times positive or negative comments occurred during the observation. At lessons, the TT-R was collected on the teacher leading the lesson only. At meals, the TT-R was collected for both lead and assistant teachers.

*WISE fidelity measure* [35]. The instrument is rated on a 1 to 4 scale, with higher scores indicating higher levels of implementation fidelity. Each core component (Role Model, Mascot and Hands-on) is assessed with the mean of 2 items.

*Semi-Structured Qualitative Interviews*. In the spring of the school year, the analyst identified teachers who were demonstrating high WISE fidelity and positive feeding practices as well as those with low fidelity; 5 from each group were randomly selected for interviews. In addition, 5 staff members (i.e., site champions and directors) were interviewed. The semi-structured, open-ended interview guide was designed to solicit feedback on the de-implementation approach and its integration with the implementation strategies, as well as fit with the context. An expert in qualitative inquiry with extensive experience conducting qualitative interviews and external to the implementation team conducted the interviews to increase the comfort-level and openness of the participants. Verbal consent was captured through audio recording. The interviews were scheduled for up to 90 min and lasted an hour on average.

### 2.6. Data Analysis

Descriptive statistics summarized teachers’ demographics. Next, we combined data from this study and the implementation trial to compare the three conditions ([1] B, [2] E, and [3] D + E). Between-group comparisons were investigated for treatment effects among the three conditions at both baseline and follow-up. Next, we investigated variance in outcomes, examined confidence intervals, and assessed the presence of practically relevant effects using paired sample t-test and ANOVA [46,47,48]. Effect sizes were examined with Hedge’s g as the sample size for the current study was around or below 20 for most groups. A small effect was categorized as between 0.2 and 0.5, while a medium effect was between 0.5 and 0.8, and a large effect was higher than 0.8 [49]. Within-group differences were examined to compare changes between baseline and follow-up time points for the D + E group.

For qualitative analysis, all transcripts were transcribed verbatim. Analyses of the qualitative data followed a pragmatic, directed content analysis approach [50,51]. The initial phase of coding focused on identifying barriers and facilitators to the utility of each de-implementation strategy. First, the PI, Research Associate, and Research Assistant met to build the initial codebook, coded three interviews to inform refinements to the codebook, and resolved questions pertaining to coding rules. Next, the Research Associate and Research Assistant coded the 12 remaining interviews independently, meeting weekly to review codes, discuss and code unclear sections of text, and direct unresolved issues to the PI.

The next phase of coding focused on examining themes within identified barriers and facilitators for the de-implementation strategies. To accomplish this goal, a primary coder completed an immersive reading phase of all the quotes in a section (e.g., barriers to peer learning collaborative). Next, the primary coder assigned thematic codes and applied these to relevant quotes, expanding the initial codebook by defining the themes and identifying key quotes. The primary coder shared these theme definitions and examples with a secondary coder who independently applied the themes to the same sections of text. Finally, all three coders met to discuss application of the themes, to identify and resolve disagreements, and to come to consensus on salience and key examples of the themes. This consensus process was used to increase trustworthiness of codes [52]. The coders repeated this process within each barrier/facilitator section for each implementation strategy.

## 3. Results

Sample Demographics. Table 2 summarizes characteristics of teachers from the three study groups. No statistically significant differences were observed among the three study groups. Overall, teachers were mostly female (99.2%) and Black (78%); a small portion of teachers were Latina (4.2%). The majority were over 41 years old (61.9%) with 11–20 years of teaching experience (38.1%). About one-third of the teachers had a bachelor’s degree (34.7%). At baseline, the weighted average mealtime lengths were 23 min for D + E, 27 min for the E group, and 27 min for the B group. At follow-up, the weighted average mealtime lengths are 24.6 min for D + E, 27.5 min for the E group, and 26.8 min for the B group. There were no significant differences between groups at baseline on outcome variables.

Delivery of Strategies to De-implementation Group. The delivery of implementation strategies to the E group was reported prior. Delivery of implementation and de-implementation strategies to the D + E group reflected planned activities for the time period of the study and as outlined in Table 1. In total, co-teacher pairs completed 90% of planned discussion prompts; peer teacher pairs completed 92% of planned discussion prompts. Classrooms earned an average of 1.7 incentives (Min = 0, Max = 4). In total, classrooms received 68 environmental reminders (*M* = 2.6), 38 educational expert recommendations (*M* = 1.5), 70 educational how-to resources (*M* = 2.7), and 32 audit and feedback reports (*M* = 1.2). Overall, coaches spent 52 h and 28 min in the field. Coaches spent 40% of their time engaging in preparation and planning. The next area where coaches spent most of their time was engaging in audit and feedback (33.3%), followed by teacher/staff engagement (12.3%), education (8.6%), other (4.9%), and assessment (1.2%).The peer learning collaborative sessions with a peer outside their classroom occurred once and took an average of 14.7 min; sessions with their classroom co-teacher occurred three times for an average of 18.6 min. Thus, meeting time in total was approximately 70 min for the school year.

Table Talk at Lessons. For D + E, Positive Comments at lessons increased from baseline to follow-up (*t*(20) = 2.314, *p* = 0.031). See between- and within-group overall differences in Table 3 and item-level differences in Table 4. Negative Comments decreased from baseline to follow-up (*t*(19) = 2.842, *p* = 0.01). For between-group comparisons, the average Positive Comments score at follow-up lessons was highest for the D + E group. The effect size between group differences for D + E and B groups was 0.34 (small), and the effect size between D + E and E groups was 0.60 (medium). The average Negative Comment scores were similar among groups (i.e., small effects).

Table Talk at Meals. From baseline to follow-up, Negative Comments for lead teachers at mealtime decreased in the D + E group (*t*(22) = 2.73, *p* = 0.01). See item-level changes in Table 4. For Positive Comments, Hedges’ *g* comparing means between groups were small with the exception of assistant teachers in the D + E and B groups which was medium (0.72) and assistant teachers in the E group and B group, which was high (0.90). For Negative Comments, effect size for between-group differences for the D + E and B groups was 0.74 (large) for lead teachers and 0.48 (medium) for assistant teachers. Effect-size differences between the D + E and E groups as well as the E and B groups were small.

WISE Fidelity. From baseline to follow-up for D + E Support, mascot use increased (*t*(21) = 3.899, *p* = 0.001). Role modeling and use of small groups did not change between time points significantly for the D + E group. Within- and between-group means for WISE Fidelity scores are presented in Table 3.

Comparing the three groups at follow-up, effect sizes for mean differences on Role Modeling between D + E and B groups and D + E and E groups were 1.34 (large) and 0.09 (<small), respectively. For Mascot fidelity, a medium effect size was found for the mean differences between the D + E and B group (Hedges’ *g* = 0.67), while the effect size between D + E and E groups was small (Hedges’ *g* = 0.09). For Hands-on fidelity, Hedges’ *g* for mean differences was small comparing D + E to both E and B groups.

### 3.1. Perceptions of De-Implementation and Implementation Strategy Combination

Table 5 provides exemplar quotes of the primary barriers and facilitators to each de-implementation strategy from interviews with participants in the D + E group. These first-hand accounts provide critical insight into key areas for sustainment and areas for improvement for future iterations of this work.

The training was based on improvisation methods to increase engagement and memorability of key messages. All feedback on the training was positive. A key facilitator was the use of role play scenarios to give teachers examples of applying the concepts in their classrooms.

The foci of the peer learning collaborative included reducing then replacing detrimental feeding practices in the context of social partnerships. The primary facilitators to the peer learning collaborative were providing opportunities for self-reflection and tailored tips and resources. Primary barriers to the peer learning collaborative included the extra work it created for the teachers and finding time to engage. Suggested improvements were automated reminders and use of a digital platform to receive resources and communicate with their coach.

Facilitation was an overarching strategy used to support both implementation and de-implementation. Primary facilitators included a positive rapport with the WISE Coach and perceptions that the coaches’ feedback was helpful. A primary barrier was the timing of the in-person visits, suggesting the desire for coaches to better coordinate the scheduling of the visits with teachers.

Audit and feedback reports were provided to teachers upon request and gave specific feedback based on their personal classroom behaviors relative to their WISE Words goals. A primary facilitator to the audit feedback reports was the perceived usefulness of the tips and strategies included in the reports. Primary barriers were lack of awareness of receiving personalized feedback and feeling it was “hard to hear”, albeit helpful, when negative feedback was shared.

Environmental reminders were provided to teachers in poster format to support keeping goals and related strategies at the front of one’s mind during meals and snacks. Primary facilitators were that the posters were used as designed and perceived as supportive of goals. One primary barrier was requests for improving the size and format of the posters.

Videos and handouts were educational resources developed to support de-implementation. For educational handouts, perceived benefit and use were primary facilitators (See Table 5). Similarly, videos were perceived to have helpful tips and resources, a primary facilitator to their use. Not using or remembering to use was a primary barrier for both the handouts and videos.

Participants were asked to reflect on their experience receiving the implementation and de-implementation supports compared to their experience doing WISE with only basic implementation strategies (training, reminders, and quality monitoring only). Primary facilitators included that the implementation and de-implementation strategies provided the teachers with new/ideas and knowledge to apply to WISE lessons. In addition, several felt that the support was an improvement over prior years. Despite some positive perceptions of the combination, primary barriers were the feeling that it was “too much” and “better before”. That is, there were several salient expressions that the additions were overwhelming and resulted in some teachers’ opinions of the innovation becoming more negative.

### 3.2. Key Events and Departure from Planned Protocol

Several key events affected the process and outcomes of our study [53]. These key events provided important context for interpretation of study findings and for understanding changes to the study protocol. Between study enrollment and intervention, the leadership changed at one of the two de-implementation sites. New leaders were openly unsupportive of the intervention and research activities whereas prior leaders had high buy-in and support. Interview participants shared the impact of this change in qualitative interviews. In addition, COVID-19 prevented follow-up measurement on the planned schedule; mid-point data collection became the final follow-up time point for all group comparisons. Measures of Effectiveness and ECE teachers’ AFC About Feeding Children Strategies and Beliefs were only available at baseline. Maintenance measures could not be collected as planned.

## 4. Discussion

Despite a mid-year interruption due to COVID-19, our study was able to combine mid-point quantitative comparisons with qualitative feedback to provide a strong evaluation of our de-implementation strategies. In this study, the D + E group showed within-group changes in the desired direction for indicators of Implementation (mascot fidelity and WISE lesson positive comments) and De-Implementation (WISE lesson negative comments and lead teacher mealtime negative comments). Further, in between-group (non-randomized) comparisons, the D + E group performed best for Implementation (mascot fidelity, role modeling fidelity, positive comments at lessons; positive comments at meals for leads and assistants, medium to large effect sizes). De-implementation of detrimental practices at meals was also significantly improved for assistant teachers in the D + E group compared to the E group support (medium effect). This pattern illustrates notable differences between the groups receiving B and D + E groups.

Our predefined thresholds for progressing this pilot work to a full-scale trial were based on identifying at least equal perceptions of feasibility and acceptability between groups, qualitative feedback in support of the de-implementation approach, and trends that favored the D + E group on the focal outcome of feeding practices [34]. COVID interruptions prevented comparisons of quantitative ratings of feasibility and acceptability and shifted the final endpoint for comparison of feeding practices to January/February instead of April/May. Further, the cessation of the study in February 2020 meant that the teachers received the portion of the intervention focused on their own self-selected goals but did not have time to work on researcher-suggested goals as designed for the spring semester. The researcher-suggested goals were targeted to areas where the ECE teacher needed the most improvement; self-selected goals were not limited to areas where the teacher needed to improve. In the face of these disruptions, we feel that the promising within-group shifts on total positive comments at lessons, negative comments at lessons, and negative comments for lead teachers at meals support the potential effectiveness of our strategy. Between-group comparisons on the focal outcome also support its promise for use of positive comments at lessons and positive comments at meals for assistants. Thus, on the whole, our study provides preliminary support for our multi-faceted, partner-selected package of de-implementation strategies. This is consistent with reviews of de-implementation strategies in clinical settings that suggest multi-faceted approaches [54] with educational components [54,55] to be most effective for de-implementation.

This study is the first, to our knowledge, to test de-implementation strategies in an ECE setting. Given that there is “no strong culture in education of having science directly influence the acquisition and elimination of practices” ([56] p. 92), and that this is especially true in the ECE environment, our study supports the potential of implementation science-informed approaches to shift negative or detrimental practices towards evidence-based ones in this setting. However, consistent with the Theory of Risk Aversion [57,58], absolute reduction of the detrimental practices proved more challenging than replacing with new practices. Our approach to de-implementation sought to support ECE teachers’ change in behavior by providing new, actionable information and surround it with social support and accountability to change. In particular, the foundation of the training in medical improv [44] allowed us to deploy exercises designed to improve communication and teamwork and create new cognitive patterns, a particularly novel approach to training ECE staff. In our case, the targeted practices to reduce/replace were challenging given that they do not require resources of ECE teachers or the ECE system (i.e., available at no cost), they are deeply engrained in cultural and ECE educational practices, have been used for many years, and seem intuitive to achieve their feeding goals (e.g., make sure children do not go home hungry). Each of these factors alone can contribute to hesitancy to change a practice [59], and they highlight how de-implementation of inappropriate practices in community-based settings may be different than in clinical settings. Our pilot also suggests that there may be differences in changing teacher practices at lessons versus mealtime settings. Future large-scale studies could explore this possibility as well as potential mechanisms underlying any differences in behavior across these contexts.

Our work captured community (rather than clinical) perceptions of de-implementation to identify important factors that influenced the process of removal/reduction as well as “what did not work”, a critical approach to advance de-implementation science [60]. Overall, qualitative feedback supported that the de-implementation strategies created needed opportunities for self-reflection, tailored resources that were helpful, and a supportive social environment for change. These key supporting factors are consistent with the conceptualization of de-implementation as deeply situated in social contexts [60] and suggest that key mechanisms of de-implementation of detrimental feeding practices may be social in nature. In particular, qualitative feedback suggests that the peer learning collaborative may create trust needed to support changes in teachers’ beliefs and self-efficacy that would precede behavior change in the classroom. Feedback from teachers also suggested opportunities for improvement of our strategies and considerations for combining implementation and de-implementation approaches (e.g., aligning format of reminders with teacher preferences/needs and removing incentives). Further, the overwhelm of some teachers suggests that there may be benefit from separating the delivery of the de-implementation and implementation strategies and/or streamlining and increasing the flexibility of the de-implementation strategies to require less time (e.g., a virtual approach). Data suggest that using coaches to remind teachers of the resources provided may be helpful.

This study had both limitations and strengths. Originally, we planned to use the Food Intake module of the Building Mealtime Environment Rating Scale [20] (BMER) to assess change in mealtime environment over time. However, all classrooms in the D + E group condition had meals in a cafeteria setting where children were served a pre-plated lunch. In accordance with center policy for the cafeteria lunch, adults decided how much food was placed on a child’s plate, and children did not serve themselves. These setting-specific practices lead all classrooms to receive a score of “inadequate” or “minimal” practice on the BMER module, which precludes scoring other practices per BMER instructions. This lack of variability and changeability (given center policy) limited the utility of the BMER for our study. Future research studies seeking to use the BMER may benefit from scoring all items as a checklist to obtain a continuous score rather than following original scoring guidelines to categorize practice on the whole. Another limitation of our study was the quasi-experimental, non-randomized nature of comparisons between the D + E, E, and B groups. We examined both within- and between-group differences in light of this limitation; future work would benefit from a fully randomized design focused on between-group findings. An additional limitation is the difference in the shortened time period between baseline and follow-up data collection because of the COVID-19 interruption in the de-implementation group. To address this limitation, we compared all groups at their mid-year data collection. Continued trends in the observed direction may have illustrated more robust effects for the D + E strategies.

Key strengths of the study include the use of a mixed methods approach, community partner engagement, and a rigorous strategy selection process. Collecting both qualitative and quantitative data to evaluate the de-implementation strategies allowed us to delineate potential improvements, better understand possible mechanisms (e.g., self-reflection), and generate hypotheses for future work (e.g., exploration of digital delivery). Our use of community partner engagement in the selection and design of both the implementation and de-implementation strategies reflects partner “difference in experiences and perceptions of overuse” of the detrimental feeding practices targeted in our study, which is critical when striving to improve rather than increase disparities in use of evidence-based practices in settings serving historically marginalized groups [56]. Teachers involved in the EBQI process of co-creating de-implementation strategies were present at the overall teacher training, which allowed them to speak to their experience with developing the strategies; this may have increased teacher buy-in on the whole. Finally, our alignment between known barriers to use of positive feeding practices and facilitators of detrimental feeding practices with selected strategies helps to address a gap in the de-implementation literature to date [54].

Even with disruptions from COVID-19, this study was able to compare basic implementation support (i.e., training and reminders only), a package of partner-selected implementation strategies, and the combination of partner-selected implementation and de-implementation strategies. Several outcomes show promise for the combination after 6 months of implementation. Specifically, the combination D + E group showed clear advantages for the outcome of Implementation (i.e., WISE lesson fidelity). This provides partial support for our hypothesis that de-implementation strategies for detrimental feeding practices can create space for the use of evidence-based feeding and obesity prevention practices. Combined with qualitative results on the experience of teachers in the D + E group, results suggest that sequencing the de-implementation and implementation strategy packages may improve implementation (versus deploying implementation and de-implementation strategies all at once). Examining this hypothesis and comparing effects on children between groups are promising areas for future research.

## Figures and Tables

**Table 1 nutrients-14-02861-t001:** Specification of multi-faceted de-implementation strategy.

Strategy	Actor(s)	Action	Temporality	Dose	Justification
Make Training Dynamic	Research staff train teachers.	Provide 6 h training using improvisation methods [44] to ground teachers in concepts.	At beginning of school year	One-time	Reduce barriers to new ideas; provide information to support change.
Peer Learning Collaborative with Goal Setting	Teachers with their co-teacher; Research staff provide prompts.	Pairs of co-teachers meet to discuss goals, examine/resolve barriers, and share accountability; Teachers select support they desire.	Starting at the beginning of the school year	Monthly, starting at training	Increase intention and commitment to change; generate behavioral alternatives; increase social support and norms for change.
External Facilitation	External facilitators (i.e., WISE Coaches)	Provide direct support to teachers in alignment with goals.	2 weeks after training for 1 year	Monthly or more upon request	Support environment to embed change; improving skills/knowledge and/or challenge beliefs.
Audit and Feedback	Research staff	Provide assessment of teacher practices at a recent meal in relation to their targeted practices.	Upon request	Upon request	Provide concrete information to teacher to increase awareness of current practice related to goals.
Remind Teachers	Research staff	Provide reminder of targeted practices (e.g., poster).	Upon request	Upon request	Give timely reminders.
Develop Educational Materials	Teachers receive handouts and/or videos from Research staff.	Provide tailored education; teachers can select practical “how-to” guides and/or expert recommendations.	Upon request	Upon request	Challenge inconsistent beliefs and improve knowledge and skills.

**Table 2 nutrients-14-02861-t002:** Teacher demographic characteristics.

	De-Implementation + Enhanced (*n* = 48)	Enhanced (*n* = 35)	Basic (*n* = 35)	Test Statistics (χ^2^/Fisher’s Exact)	Total (*n* = 118)
**Female, %**	97.9	100	100	1.5	99.2
**Race, %**				2.1	
White	22.9	11.4	22.9		19.5
Black	75.0	85.7	74.3		78.0
Other	2.1	2.9	2.9		2.5
**Ethnicity, %**				2.3	
Latina	2.1	2.9	8.6		4.2
**Age, %**				11.43	
19–24 years	0	14.3	2.9		5.1
25–34 years	14.6	14.3	25.7		17.8
35–40 years	18.8	11.4	14.3		15.3
41+ years	66.7	60	57.1		61.9
**Education, %**				14.0	
High School	6.3	11.4	11.4		9.3
Some College	10.4	25.7	17.1		16.9
Associate’s	29.2	37.1	31.4		32.2
Bachelor’s degree	41.7	25.7	34.3		34.7
Master’s or higher	12.5	0	2.9		5.9
Other	0	0	2.8		0.8
**Teaching experience, %**				11.1	
<1 year	0	5.7	0		1.7
1–10 years	25.0	34.3	34.3		30.5
11–20 years	52.1	25.7	31.4		38.1
21+ years	22.9	34.3	34.3		29.7

Results based on listwise deletion.

**Table 3 nutrients-14-02861-t003:** WISE lesson and mealtime fidelity: between-group differences at baseline and within-group differences across time.

	De-Implementation + Enhanced			Enhanced			Basic		
Mean (SD)	Baseline	Follow-Up	*p*-Value *	Baseline	Follow-Up	*p*-Value	Baseline	Follow-Up	*p*-Value
**WISE Lessons**	M(SD)	M(SD)		M(SD)	M(SD)		M(SD)	M(SD)	
Role Model	2.8 (0.6)	3.0 (1.1)	0.30	3.0 (0.8)	2.9 (1.2)	0.82	2.7 (1.1)	2.6 (1.0)	0.65
Mascot	2.2 (0.8)	2.8 (1.2)	0.001	2.4 (1.1)	2.8 (1.0)	0.16	2.0 (1.1)	2.1 (1.2)	0.67
Hands on	2.1 (0.7)	2.5 (1.1)	0.08	2.8 (0.8)	2.4 (1.2)	0.21	2.41 (1.2)	2.6 (0.9)	0.52
Positive Comment (Lead)	14.2 (7.6)	19.5 (11.3)	0.03	14.8 (7.6)	12.9 (1.8)	0.51	14.7 (9.3)	15.2 (7.6)	0.89
Negative Comment (Lead)	6.8 (4.0)	4.2 (3.0)	0.01	2.7 (1.7)	5.0 (4.6)	0.11	2.4 (2.1)	4.1 (2.6)	0.17
**Mealtime Fidelity**									
Positive Comment (Lead)	9.2 (5.8)	7.3 (6.1)	0.13	8.3 (4.3)	7.1 (4.3)	0.27	7.6 (6.2)	6.9 (5.6)	0.64
Negative Comment (Lead)	13.4 (5.7)	8.7 (5.0)	0.01	12.9 (8.8)	7.4 (5.2)	0.03	10.7 (4.0)	5.8 (4.3)	0.001
Positive Comment (Assist)	4.7 (5.2)	5.7 (6.5)	0.40	6.2 (3.3)	5.5 (4.6)	0.73	6.1 (3.0)	2.4 (2.3)	0.002
Negative Comment (Assist)	10.6 (6.2)	8.3 (6.0)	0.18	9.1 (5.7)	5.1 (5.0)	0.08	8.5 (3.8)	5.6 (4.8)	0.06

* Note: Within-group *p*-value is shown in the table.

**Table 4 nutrients-14-02861-t004:** Within-group table talk comments for de-implementation + enhanced group.

Lesson	Baseline	Follow-Up
**Positive Comments**.		
Positive Comments-Teacher focus	1.5 (1.1)	1.6 (1.5)
Positive Comments-Food focus	0.8 (0.6)	1.4 (1.5)
Hunger cues	0.1 (0.2)	0.2 (0.5)
Encourage trying in positive way	1.8 (1.5)	2.5 (2.6)
Nutrition coaching (focus on child experience) *	3.5 (3.1)	6.5 (5.5)
Exploring foods (focus on food itself)	6.6 (3.6)	7.4 (4.8)
Total Positive Comment *	14.2 (7.6)	19.5 (11.3)
**Negative Comments**		
Negative Comments about the food served	0.3 (1.2)	0.1 (0.2)
Pressure to eat ***	1.1 (0.9)	0.3 (0.7)
Hurries to finish eating **	0.2 (0.3)	0
Discourage manipulating food	0.1 (0.2)	0
Social Comparison **	0.2 (0.4)	0
Threats (to encourage eating)	0.03 (0.1)	0
Preference for unhealthy food *	0.1 (0.1)	0
Food as a reward *	0.1 (0.2)	0
Focus on behavioral control ^ƚ^	5.03 (3.2)	3.6 (2.9)
Total Negative Comment *	6.8 (4.0)	4.2 (3.0)
**Mealtime**		
**Positive Comments**		
Positive Comments-Teacher focus	0.9 (0.8)	1.0 (1.1)
Positive Comments-Food focus	1.0 (1.0)	0.5 (0.9)
Hunger cues	0.4 (0.6)	0.5 (1.0)
Encourage trying in positive way	2.1 (1.8)	1.6 (1.6)
Nutrition coaching (focus on child experience)	1.7 (1.3)	1.7 (2.2)
Exploring foods (focus on food itself)	3.5 (3.4)	2.2 (2.1)
Total Positive Comment	9.5 (5.9)	7.6 (6.0)
Total Positive Comment Classroom	14.6 (8.7)	14.9 (11.5)
**Negative Comments**		
Negative Comments about the food served	0	0
Pressure to eat *	4.3 (2.6)	2.5 (2.4)
Hurries to finish eating	1.0 (1.2)	0.7 (0.9)
Discourage manipulating food **	0.9 (0.9)	0.3 (0.9)
Social Comparison	0.4 (0.6)	0.2 (0.7)
Threats (to encourage eating)	0.03 (0.1)	0
Preference for unhealthy food	0.1 (0.2)	0
Food as a reward	0.1 (0.3)	0.04 (0.2)
Focus on behavioral control	6.1 (2.9)	5.2 (3.7)
Total Negative Comment	13.0 (5.5)	9.0 (4.9)
Total Negative Comment Classroom **	23.6 (8.0)	17.6 (5.7)

^ƚ^*p* < 0.01, * *p* ≤ 0.05, ** *p* < 0.01, *** *p* < 0.001; All scores are for Lead Teachers unless indicated as classroom totals.

**Table 5 nutrients-14-02861-t005:** Key facilitators and barriers by de-implementation strategies.

Strategy	Facilitators	Barriers
**Make Training Dynamic**	**Realistic Scenarios** I think they are doing a pretty good job because even like in our training, they actually act out scenarios… how we were supposed to, you know, talk and things we’re supposed to say. You know, so they do a really good example of how they want us to role model for the kids.	None noted
**Peer Learning Collaborative**	**Self-Reflection** “I do believe that it is good to identify your weaknesses and your strengths and to aim to, you know, do more things in a positive way and to chance obviously to self-evaluate. And I think it’s important to set aside time to self-evaluate, why am I doing that? A lot of what am I doing that needs to be changed, so as far as that goes…I think that was positive.” **Tailored Tips and Resources** “They actually asked us asked us like this year in the binders, like what resources do you need from us and we could tell them a specific resources, so I thought that was good.”	**Extra Work** “A lot of us have like some stuff on our plate is really just the idea of like using our break time or after school time to get together and do this. It just wasn’t realistic.” **Time** “I like the way they set it up to do with the binders and meeting with your Para and your teacher, like, that’s not a bad. Like it’s a good idea if there was time for it, you know? Like, it’s not it’s not bad like in itself is just not you know, feasible because you know, of our schedule…”
**External Facilitation**	**Positive Rapport** “You know when somebody comes in and observe it was tense, but it was never that it was the opposite. She would come in and she would do her job, but the kids would say something funny and she would laugh, you know she was just very helpful... we knew them, and we recognize them, and it was okay to have them coming in at any time.” **Helpful Feedback** “It was helpful to receive that feedback because it made me aware.”	**Timing of Visits** “She would pull me to the side and talk to me and she would tell me. We had so much going on. I had nine children with IP and they would constantly get pulled out and so some days it would be overwhelming… because when you have behavior problems and IP’s you know sometimes it’s beyond just two teachers.”
**Audit and Feedback**	**Supported Improvement** They would watch to see what we were doing and then they give us feedback. So, it, it improved.	**Did not Perceive as Feedback** If I had that pat on my back or a way to know that I’m doing a good job or what I can do it to make it better, it would be super…You only have the training once a year in august, how can you improve yourself or how could you know what you are doing wrong? **Hard to Hear** I did at one time get feedback that there was something that I was missing that I didn’t need to do every time, and that was hard to hear. But okay, and it means I can fix it in the future.
**Reminders**	**Used as Designed** And they remind us of our goal, and so that that laminated poster board that they gave me was really kind of helpful, because I had to know ‘Okay, this is my goal for this,’ you know… so that laminated poster board I think was probably the most best for me.	**Improve Format** Ok, we did get a couple of posters, …but if we could have got something more bleft and more visual, big. I could have put them in the middle of my room where I do most of my teaching for circle time. That would have been very helpful as well.
**Educational** **Hand-outs**	**Use as a Reminder** “That’s usually what I do, I pick it (handout) up and looking at it like before, you know the day before we do our Wise activity.” **Perceived benefit** “We hung them up on the wall in our feeding area, and we used them for reference while we were doing the activity… I think they were pretty good, pretty colorful and eye catching.”	**Does not Remember** “They gave us so many, so I don’t remember the exact one.” **Did Not Use** “…if somebody hands me like a paper handout…like especially for WISE I put it in that binder and honestly, kind of like out of sight out of mind or kind of forget about it.”
**Educational Videos**	**Helpful Tips and Resources** “Like I was having trouble with saying positive things about the food so much, they gave me the videos and handouts to show me exactly how to do it...Then I was more comfortable with it.”	**Did Not Use** “I didn’t get a chance to watch them.”
**Implementation and De-implementation Combination**	**New ideas/knowledge to apply to WISE** “I felt like it enlightens me to how to incorporate Wise in my classroom and as far as breaking it down on the on the level of the children on their understanding.” **Improved support** “It was just improvement. You know, just reinforcing, making sure that we were doing what we were supposed to do…. Reinforcing those words to the children. So, it was just moving a step up.”	**Too Much** “Having all the extra WISE stuff on top of it can just take the fun out of it.” **Better Before** “I think honestly that that binder kind of turned a couple of teachers this year. Like you know, everybody was cool with WISE, and then we had to get that binder in and it was extra paperwork. It was like inviting a mind change towards WISE a little bit.”

## Data Availability

De-identified data will be made available based on reasonable request to the corresponding author.

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
