# Peer review of "De-Implementation of Detrimental Feeding Practices in Childcare: Mixed Methods Evaluation of Community Partner Selected Strategies"

_nutrients, 2022, doi:10.3390/nu14142861_

Round 1

Reviewer 1 Report

This is a pilot studying effect of de-implementation of detrimental feeding practices in early care as a step prior the very implementation  of evidence based strategies aiming to prevent obesity. Research is novel and fills in a gap knowledge since not much is known on de-implementation strategies. Strengths are community-based context, stakeholders involvement and very well described and reproducible methodology including both quantitative and qualitative measurements. A solid ground for further research.

Introduction: very well described background. No comments.

Methods: very comprehensive description of methodology; I would refer to the published protocol at the very beginning of this section (first in Measures chapter in the current version);

Results: in general, it is somewhat difficult to follow results in the narrative form, I would consider presenting more results in forms of tables/figures instead. Interesting finding was the more pronounced effect of intervention  seen in teachers' approach during lessons compared to at mealtime (with even negative change in the latter). This should be preferably commented on, possible causes and strategies to affect mealtime more should be discussed. 

Specific comments: Discussion would benefit from subheading; lines 113, 133 (+more) "Blinded" instead of a reference.

Author Response

We have addressed each reviewer comment in the attached file. 

Reviewer 2 Report

Thank you for the opportunity to review this exciting manuscript. Your investigation of de-implementation of detrimental feeding practices is an important contribution to nutrition science, and serves as an example for other nutrition researchers on how to do robust implementation research. This is a complex topic and while this paper is dense, it is also accessible. As both a nutrition scientist and a parent with a kid in childcare, I truly enjoyed reading it.

I have one major concern, and a few moderate/minor concerns, but overall, I’m very impressed.

My major concern relates to the alignment of the qualitative investigation with the paper objectives. First, in general, since your goal with this paper is to describe the de-implementation strategies and it seems that you only did interviews with the D&E group, what is the justification for including the barriers and facilitators for the implementation strategies? I just don’t think the set up is there for this to align with your overall objective. For example, I have no idea what champions, cutting board or incentives are. I am guessing they help explain the WISE fidelity score? But I’m not really sure. And I’m unsure how it relates to the overall goal of the paper. Second, of the six de-imp strategies you describes in Table 1, I only see one discussed in the interviews? Third, in the section about imp and de-imp strategies combined, I thing these strategies are great but as far as I can tell they are not discussed in the methods. I do appreciate the rich description, particularly the reporting of discrepant results.

Moderate concerns

·        The use of so many different terms for the people involved in the process: ECE teacher, educator, coach/implementation agent, champion, improvisational trainer, etc. While you do an excellent job differentiating the intervention from the imp/de-imp strategies, the cast of characters is confusing. I wonder if a figure would help? Or maybe just some extra in the “participants” section? Several of these groups are not well-described, so could they be combined? And in particular, Champion is such an important role and loaded word in imp sci, but it is not really defined in this paper at all.

·        Pg 5, Line 189. I think a few more details on the TT-R are needed, unless I’m missing something. How was TT-R administered? Was it an independent observer or self-report? Was it one day, one week, a few different periods? And how long after the training de-implementation strategies took place? What was the total score possible? What’s a “good score”? And is the score reported in the results indicating the # of positive and negative comments?

Minor:

·        In the intro when discussing negative outcomes of detrimental feeding practices (Lines 70-73), are the papers references (refs 20, 24, 26-29) specific to childcare-aged children?

·        The last paragraph of the intro needs some clarity. For the study goals, would it make more sense to describe between group then within? You describe it in this order in the analysis and I like that better. If not, prior to the sentence I would define the trial groups. Also, the “Intervention” section of the Methods is the first introduction of WISE. So I would recommend  adding a bit about WISE as the EBI, then the trial to test imp/de-imp strategies for WISE (or non-WISE)

·        Line 107 – you have not yet defined EQBI meetings, so maybe just move this incentive note down to 2.4. What is the difference between educators who were surveyed and participants who were interviewed?

·        A few questions about the EQBI. I marveled at this rich description and exemplary application of implementation science methods. I almost feel like it should be its own paper, or perhaps a more detailed supplemental file/field guide.

o   Are participants in EQBI also receiving the imp/de-imp strategies? What influence might that have?

o   Can you define “improvisation leaders”? Were these trained members of the study team or external people?

o   Were the EQBI sessions sequential? And over what period of time did they occur? And then how soon after were the de-imp strategies rolled out to the D+E centers? Was it in tandem with the imp strategies? And how long after that was the follow up evaluation? Apologies if I missed these details. compared to evaluation?

·        Table 1 is awesome but it doesn’t quite align with the narrative. I think I mostly followed but I might better link the narrative. Like “First, dynamic training... Then, Peer learning collaborative..”

·        In the qualitative findings and discussion, I’d love to see you come back more to the point made in the introduction about teachers’ personal beliefs and self-efficacy. You obviously got at least some to buy in to the EQBI process, so some people bought in. Did you face pushback? How was it dealt with? Did people come around?

·        As written, the quant results are a lot to take in. Especially since they are repetitious of the table, could it be streamlined? One thing that might help is to make the headers “Change in Table Talk” so that you can eliminate the “from baseline to followup”. I also think you could refer to the table for the descriptives, and including only the inferentials in the narrative. But defer to you, and to the editorial team, to determine the best course of action there.

·        The part in the discussion about covid (pg 14, Lines 406-417) seems like it belong in the “key events” section of the results. Or in the methods somewhere. The mention of self-selected goals vs research-selected goals came kind of out of the blue for me.

Very minor:

·        Page 3, Line 126 – I see EBQI here. Would either add more clarification above, or just leave this here and indicate the incentive.

·        Page 5, Line 193 – With so many acronyms in this paper and this one only mentioned once, I would remove this and just go with “ECE teacher” that you’ve used above.

·        Table 3: you have the first column labeled as WISE Lesson Fidelity and Mealtime Fidelity but I feel like what you are describing, and how the narrative is labeled, is Table Talk. In the methods you do not have a section to describe mealtime fidelity, and WISE fidelity (role model, mascot, hands on) should be separate from positive and negative comments

·        Pg 7, Lines 277-278 There are a few typos here

·        Pg 9, Line 306 missing a period

·        Pg 9, Lines 310-324 is WISE fidelity, not Table Talk? I think a subheader may have accidentally been deleted based on the accidental bracket

·        Table 5, In Cutting Board facilitators quotes, does Windy need to be de-identified?

Author Response

Please see attachement. 
